# Mortality-Associated Factors in a Traumatic Brain Injury Population in Mexico

**DOI:** 10.3390/biomedicines12092037

**Published:** 2024-09-07

**Authors:** Erick Martínez-Herrera, Evelyn Galindo-Oseguera, Juan Castillo-Cruz, Claudia Erika Fuentes-Venado, Gilberto Adrián Gasca-López, Claudia C. Calzada-Mendoza, Esther Ocharan-Hernández, Carlos Alberto Zúñiga-Cruz, Eunice D. Farfán-García, Alfredo Arellano-Ramírez, Rodolfo Pinto-Almazán

**Affiliations:** 1Sección de Estudios de Posgrado e Investigación, Escuela Superior de Medicina, Instituto Politécnico Nacional, Plan de San Luis y Díaz Mirón, Ciudad de Mexico 11340, Mexico; erickmartinez_69@hotmail.com (E.M.-H.); juancast0508@gmail.com (J.C.-C.); cefvenado@hotmail.com (C.E.F.-V.); cccalzadam@yahoo.com.mx (C.C.C.-M.); estherocharan@hotmail.com (E.O.-H.); charlitos79@hotmail.com (C.A.Z.-C.); 2Fundación Vithas, Grupo Hospitalario Vithas, 28043 Madrid, Spain; 3Efficiency, Quality, and Costs in Health Services Research Group (EFISALUD), Galicia Sur Health Research Institute (IIS Galicia Sur), SERGAS-UVIGO, 36213 Vigo, Spain; 4Maestría en Ciencias de la Salud, Escuela Superior de Medicina, Instituto Politécnico Nacional, Plan de San Luis y Díaz Mirón, Ciudad de Mexico 11340, Mexico; dr.evelyngalindooseguera@gmail.com; 5Servicio de Medicina Física y Rehabilitación, Hospital General de Zona No 197, Texcoco 56108, Mexico; 6Dirección Médica, Hospital Regional de Alta Especialidad de Ixtapaluca, Ixtapaluca 56530, Mexico; gadriangasca1@yahoo.com.mx; 7Laboratorio de Bioquímica, Sección de Estudios de Posgrado e Investigación, Escuela Superior de Medicina, Instituto Politécnico Nacional, Plan de San Luis y Díaz Mirón, Ciudad de Mexico 11340, Mexico; efarfang@ipn.mx; 8Jefatura de Terapia Intensiva, Hospital Regional de Alta Especialidad de Ixtapaluca, Ixtapaluca 56530, Mexico

**Keywords:** traumatic brain injury, seizures, aggregate limb trauma, diabetes mellitus, heart disease

## Abstract

Background: Traumatic brain injury (TBI) is a major cause of death and disability, with a rising incidence in recent years. Factors such as age, sex, hypotension, low score on the Glasgow Coma Scale, use of invasive mechanical ventilation and vasopressors, etc., have been associated with mortality caused by TBI. The aim of this study was to identify the clinical and sociodemographic characteristics that influence the mortality or survival of patients with TBI in a tertiary care hospital in Mexico. Methods: A sample of 94 patients aged 18 years or older, from both sexes, with an admitting diagnosis of mild-to-severe head trauma, with initial prehospital treatment, was taken. Data were extracted from the Single Registry of Patients with TBI at the Ixtapaluca Regional High Specialty Hospital (HRAEI). Normality tests were used to decide on the corresponding statistical analysis. Results: No factors associated with mortality were found; however, survival analysis showed that the presence of seizures, aggregate limb trauma, and subjects with diabetes mellitus, heart disease or patients with four concomitant comorbidities had 100% mortality. In addition, having seizures in the prehospital setting increased the risk of mortality four times. Although they did not have a direct association with mortality, they significantly decreased survival. A larger sample size is probably required to obtain an association with mortality. Conclusions: These results reflect the severity of the clinical situation in this population and, although no risk factors were identified, they enlighten us about the conditions presented by patients who died.

## 1. Introduction

Traumatic brain injury (TBI) is an alteration in brain function, temporary or permanent, caused by an external force, such as traffic accidents (car, bicycle, motorcycle, injured pedestrian), falls, assault with physical force or weapons, blast wave injury or sports injury, which can lead to disability or death [1,2,3,4]. According to the Glasgow Coma Scale (GCS), TBI can be classified as mild (GCS 14–15), moderate (GCS 13 to 9) and severe (GCS less than 9), considering the time of loss of alertness, post-traumatic amnesia and clinical data of acute neurological deficit [1,5,6]. 

In the last 3 years, injuries have increased mostly in low- and middle-income countries [7,8]. Among the causes, according to the World Health Organization (WHO), a quarter of TBI-related deaths are due to suicides and homicides, with another quarter resulting from traffic accidents [7]. On the American continent, traffic accidents are the leading cause of TBI-related deaths in young adults (aged 15 to 29 years), and no significant difference in incidence according to sex has been found [7,9]. In Mexico, however, according to a 2022 report from the National Institute of Statistics and Geography (INEGI), deaths due to traffic accidents and violence were placed in seventh and eighth, respectively, (with traffic accidents being the leading cause of TBI-related deaths in the country) [8]. Additionally, for individuals who survive TBI events, patients often suffer from lifelong TBI-related impairments, with one report suggesting that 16% of TBI-survivors develop a permanent disability [7,9,10]. Nowadays, neuromonitoring strategies have been developed to identify neurophysiological worsening that may indicate a new or ongoing secondary lesion [11,12,13,14,15,16,17,18,19,20]. 

Epidemiological studies have reported the importance of identifying the deficiencies and strengths in therapeutic and preventive strategies, allowing them to be optimized and thus reduce the health impact, both in terms of costs and in the disabilities generated [9,10]. Therefore, the aim of the present study was to identify the clinical and sociodemographic characteristics that influence the mortality or survival of patients with TBI in a tertiary care hospital in Mexico.

## 2. Materials and Methods

In the study, the sample of 94 patients comprised men and women aged 18 years and older, and was determined with the following inclusion criteria: admitting diagnosis of mild-to-severe head trauma with initial prehospital treatment (use of anticonvulsants, crystal solutions or colloids, and mannitol), directly hospitalized from the emergency department or referred from other hospital units, with informed consent upon admission to hospital and from the emergency department, which included authorization for the observational study protocol between February and October 2023. The exclusion criteria were polytrauma patients without head trauma. The elimination criteria were patients whose outcome was unknown, either by voluntary discharge or transfer to another unit. The data were extracted from the electronic clinician-reported medical records of the Ixtapaluca Regional High Specialty Hospital (HRAEI) and captured in an electronic platform called the Single Registry of Patients with TBI created by the HRAEI planning service. The study was conducted in accordance with the Declaration of Helsinki and approved by the Research and Research Ethics Committees of the HRAEI (protocol code NR-32-2021 and NR-CEI-HRAEI-32-2021).

This is an observational, longitudinal, retrospective cohort study. The statistical analysis was carried out with the IBM SPSS version 25 program. Normality tests such as the Kolmogorov–Smirnov, asymmetry, kurtosis, histogram and q–q graphs were performed to decide the corresponding statistical tests. For data with a normal distribution, Student’s t-test and Pearson’s test were applied, and for data with a non-normal distribution, we used the Mann–Whitney U, Chi-square, Spearman’s test and risk estimation by odds ratio which was obtained through bivariate and multivariate logistic regression models. The survival curve and risk estimation were calculated using the Kaplan–Meier models.

## 3. Results

A sample of 94 subjects with trauma was obtained, of which 82 (87.2%; 95% CI = 0.77–0.92) were men and 12 (12.7%; 95% CI = 0.07–0.22) were women. The average age was 34.8 + 15.3 years for men and 57.3 + 12.7 years for women. An average weight for men of 79 + 13.1 kg and 73.3 + 19.6 kg for women was also reported. Regarding the clinical characteristics of admission, in terms of severity, 22.34% of the patients had mild TBI, 22.34% had moderate TBI and 55.31% had severe TBI, whereas the mechanism of trauma was presented as follows: car accidents at 46.9% (these included frontal impacts (39.6%), rollovers (16.7%), side impacts (9.4%), rear-end impacts (5.2%) and unknown mechanisms (29.1%)), falls 32.3%, aggression by third parties at 13.5%, injured pedestrians at 5.2% and unknown mechanisms at 2.1%. In addition, it was observed that patients had an average GCS of 9 + 3.6, and an average systolic blood pressure (SBP) and diastolic blood pressure (DBP) of 126.7 + 25.8 mm Hg and 72.6 + 14.2 mm Hg, respectively. Moreover, 23% (95% CI = 0.15–0.33) of the subjects were under the influence of alcohol at the time of injury and 4.3% (95% CI = 0.01–0.11) were under the influence of some illegal drugs during the accident. The average travel time from the accident site to the hospital was 37.8 min (95% CI = 27.3–48.2). The mean serum hemoglobin (Sh) was 13.2 g/dL (95% CI = 12.6–13.8) and the mean serum creatinine (Sc) was 1.02 mg/dL (95% CI = 0.9–1.1). During hospitalization, the average stays lasted 31.4 days (95% CI = 25.1–37.5), with a mortality rate of 19.6% (95% CI = 0.69–0.87), and the average dose of vasopressors used was 0.15 μg/kg/min (95% CI = 0.09–0.21) (Table 1). 

Regarding the demographic characteristics of the patients, 34.4% completed primary school, 35.6% completed secondary school, 17.2% obtained a high school diploma, 4.6% had a bachelor’s degree, 3.4% were illiterate and 4.6% were unknown. The nutritional status was distributed in the following descending order: overweight (40.9%), normal weight (36.3%), obesity grade I (18.1%), obesity grade II (3.4%) and underweight (1.1%). The socioeconomic level is defined by the classification used by the Mexican Ministry of Health, which contains 7 levels: level 1x is the stratum with the least resources and level 7 is with the greatest resources. In this study, the sample was mostly made up from the lowest socioeconomic strata, where level 1 represents 71.2% (95% CI = 59.88–80.54), level 2, 22.5% (95% CI = 14.22–33.46), level 3, 1.2% (95% CI = 0.06–7.72), level 4, 2.5% (95% CI = 0.43–9.57) and level 5, 2.5% (95% CI = 0.43–9.57). 

When studying whether there is an association between mortality and the demographic and clinical characteristics of patients with TBI, no significant differences were observed for age, even though patients who died were an average of 3.5 years older than those who lived (deceased = 40.5 + 15.23 years; alive 36.9 + 15.23 years). The comparison of weight and SBP was similar in both groups; however, DBP was shown to be higher in subjects who died than in those who survived (deceased = 76 + 11.09 mm Hg; alive = 71.7 + 15.04 mm Hg). No statistically significant differences were observed from the GCS, vasopressor dose used, hemoglobin and creatinine values. Hospital stay was significantly longer for living subjects than for those who deceased (alive = 33.1 + 30.54 days; deceased = 21.7 + 27.9 days) (Table 2). 

Regarding the association of the clinical status of traumatized subjects with mortality, only age and DBP could be risk factors. In this sense, we identified that individuals who were subjected to mechanical ventilation or those who were administered vasopressors could reduce the risk of mortality by half. In contrast, patients who were under the influence of alcohol at the time of injury with anisocoric pupils or with diffuse lesion type III could increase the risk of a fatal outcome (Table 3). 

The overall survival for subjects with TBI was 44% (95% CI = 0.1–1.0), and half of the traumatized subjects died within 120 days of hospital stay. Likewise, subjects who reached the end of the 187-day follow-up still had a cumulative mortality risk of 73% (Figure 1). 

Following further analysis, brain death was registered in 9%, and the survival was 42.1% in non-brain-dead subjects (95% CI = 0.1–1.0), in contrast to brain-dead subjects who reached 100% mortality at 24 days of hospitalization (1.5 times higher cumulative risk for a fatal outcome). According to sex, the approximate survival rate was 50% for both sexes, but it should be noted that the median survival of women was 20 days while men recorded 120 days. Regarding the analysis for aggregate trauma, it showed results close to statistical significance (*p* = 0.07). For these subjects, the highest risk group corresponded to patients with limb trauma, achieving 100% mortality. Subjects with trauma to the thorax (95% CI = 0.1-1.0), neck (95% CI = 0.4–1.0), and neck and spine (95% CI = 0.4–0.9) showed survival values of 45.8%, 68.6% and 61.8%, respectively. On the other hand, mortality in patients without added trauma survival was 41.1% (95% CI = 0.1–1.0) (Figure 2 and Table 4). 

During the analysis of the survival curves, statistically significant differences were reported in those who had seizures (survival = 0%) compared to those who did not (survival = 32%; 95% CI = 0.07–1.0). The hospital service where the use of vasopressors was initiated was also statistically significant, as it was observed that in the emergency room mortality was 100%, while in other areas of care, such as the operating room, mortality appeared in 66.7% of the cases (95% CI = 0.3–1), ICU in 54.3% (95% CI = 0.29–0.98) and ITU in 50% (95% CI = 0.18–1.0). In relation to patients who did not require the use of vasopressors, the survival was 100%. Regarding the personal pathological background, subjects with diabetes mellitus (DM), heart disease (CD) or the group of patients with four comorbidities (DM, systemic arterial hypertension (SAH), pulmonary hypertension (PH) and congestive heart failure (CHF)) achieved a mortality of 100%. Patients who presented either SAH (95% CI = 0.18–1.0), DM in conjunction with SAH (95% CI = 0.21–1.0) or those who denied previous diseases (95% CI = 0.24–1.0) reported an approximate survival rate of 50% (50%, 53.3% and 55.8%, respectively) (Figure 2). 

Interestingly, in the case of seizures, despite not being statistically significant, they showed a 4-fold increased risk of mortality. These results were previously adjusted for age (Table 5). 

## 4. Discussion

In the present study, no risk or protective factors were identified associated with mortality in TBI. However, the survival analysis showed that seizures and the need for vasopressor use in the emergency room may be related to increased mortality since they are absent in polytrauma patients who survived. The presence of preexisting comorbidities to the trauma, although it had a low incidence, does have an impact on survival.

This study reported an age range of 34 to 41 years, with a higher percentage of men but with women reaching the survival curve in less time. Accordingly, several reports mention the same age range of our study, being between 26.7 and 49.5 years [21,22,23,24,25]. On the other hand, the WHO has declared that men have twice the risk of mortality due to accidents [7]. Numerous authors have associated mortality in TBI with sex and age [6,22,23,26,27]; however, this was not observed in the present study, probably due to the sample size. Other studies have reported that postmenopausal women have a worse clinical prognosis after TBI [28,29], which is associated with the depletion of sex hormones [30,31,32]. 

In regards to the socioeconomic status, according to the WHO, deaths from injuries occur 2.5 times more frequently in poor European countries, and more than 90% of these happen in low-income countries [7]. In the present study, more than 50% of the subjects were low-income people (level 1 of socioeconomic status). Various studies have mentioned that there is an increased risk of TBI due to less safe conditions, fewer prevention activities and even limited access to medical and rehabilitation care due to the low-income status [3,7,33,34,35,36,37]. Although an association with mortality was not obtained in this study, the high prevalence of low socioeconomic status could be confirmed. 

According to the magnitude of the trauma, it was observed in this study that severe TBI had the highest prevalence in more than 50% of the cases, mainly caused by car accidents, specifically motorcycle accidents. Interestingly, Nguyen et al., in their meta-analysis of 4944 studies from different continents, reported that mild TBI was the most prevalent without mentioning the most common cause [6]. It is known that the severity of the injury directly influences the prognosis, use of health resources, functional recovery and rehabilitation needs after the injury, and is directly related to mortality [38]. In concordance with our results, Capizzi et al. in the USA and, at the same time, Wu et al. in China described that the main cause of TBI was traffic or automobile accidents, these being classified between moderate and severe stages depending on their clinical frame [21,24]. Nevertheless, other authors such as Mauritz et al., Brazinova et al. and Giner et al. reported that at the European continent the main cause of TBI were falls [7,22,23,24]. Remarkably, Giner et al. reported that in Spain, severe TBI was more prevalent in adults over 50 years of age and was also associated with falls secondary to the use of anticoagulants [24]. Although no association was found between the severity and the cause of the trauma in this study, the international literature has consistently reported it, so we encourage to continue this type of research in the Mexican population.

Severe trauma is caused by high-energy mechanisms, and in many cases, more than one injury occurs. In the present study, it was observed that aggregate trauma had a low incidence; however, when performing the analysis with survival, it was found that limb trauma has a greater relationship with mortality, while other types of traumas only reduce survival. Although, several authors have reported the prevalence and incidence of TBI; the analysis according to mortality or survival is not routinely performed. Lafta et al., conducting a retrospective study of 469 patients in Iraq, identified that in those with TBI, the associated injuries were facial injuries (28.4%), skull fractures (32%), thoracolumbar fractures (10.9%), limb injuries (23.9%), abdominal injuries (6.8%) and chest injuries (6%) [38]. Tolonen et al. and Macciocchi et al. described that patients with severe TBI also had spinal cord injury, with a prevalence of 74% and 60%, respectively [39,40]. Similarly, Leijdesdorff et al., in his multicenter study of 6061 patients, observed that injuries related to TBI were pulmonary contusions and rib fractures and, to a lesser extent, lower extremity trauma and face trauma [41].

The literature evidences that the influence of alcohol and illegal drugs at time of injury increases the likelihood of suffering all types of injuries, including TBI [41,42,43,44]. In the present study, the consumption of these substances was underreported since some of the patients were unconscious at admission and the data remained unknown. In addition, due to the severity of the injuries, in many cases, it was not possible to determine this parameter during admission and others were referred from other hospitals where this information could not be verified. These factors may have influenced the lack of relationship with mortality or survival of the patients in the present study. In their reports, Moore E.E. and VanderVeen J.D. informed that in hospitalized patients, the influence of drugs prior to the injury and/or during the accident occurred in 50% of cases, and, interestingly, consumption decreased after the trauma [41,42]. Contrary to this, Leijdesdorff et al. and Madan et al. found that the higher the blood concentration of alcohol, the fewer brain injuries, skeletal injuries and concussions the patients had, and even better survival [43,44]. 

Pre-trauma diseases predispose to a worse prognosis and increase mortality. Diverse authors reported that patients with TBI may have one or more pre-existing comorbidities at the time of injury. These comorbidities may increase the risk of mortality and even development of chronic diseases at long term, regardless of the time of onset of the disease, which may result in overall delays in rehabilitation progress [45,46]. Within this research, it was observed that the presence of comorbidities significantly reduced survival. The presence of heart disease, diabetes mellitus and multiple comorbidities increased mortality by 100%, even in this population who were considered young adults. In accordance, Hammond et al., in a retrospective study, indicated that the most common comorbidities in patients with TBI were pre-existing hypertension, dyslipidemia and anxiety, whereas diabetes mellitus was developed after the trauma. Furthermore, it was later associated with chronic pain, neuroendocrine dysfunction, fatigue, sleep disturbances, urinary disorders, new-onset stroke and post-traumatic seizures or epilepsy [47]. 

The clinical data in traumatized subjects that can enlighten us about the severity of the injury are the GCS score, pupillary abnormalities and blood pressure, specifically systolic blood pressure (SBP) and hypotension, which may be present from hospital admission and have been associated with mortality [12,16,23,36,48,49,50,51,52]. In this study, only diastolic blood pressure (DBP) was close to significance, and the mean of this was not found within hypotension at hospital admission, although it did occur in the following hours. Statistical significance was not observed for the GCS score and pupillary abnormalities.

TBI by itself represents a high risk of mortality, not to mention the possible short-, medium- and long-term complications that can develop; therefore, various neuromonitoring strategies should be implemented. As for clinical signs, the decrease in ECG, the presence of new focused motor deficits, decrease in or loss of pupillary reactivity, pupillary asymmetry, appearance of herniation syndrome, alterations in blood pressure, heart rate and body temperature, imaging methods (skull tomography, magnetic resonance imaging or electroencephalogram) and ICP monitoring should be taken into account. Thus, the use of vasopressors (norepinephrine, vasopressin, etc.), the dose and the type, as auxiliary to maintain adequate organ perfusion, indicate a high risk of mortality [53,54,55]. In this research, although no association was found with mortality, it was observed that patients who did not require vasopressors during the hospital stay had a 100% survival. In addition, the time of initiation of these drugs influenced survival; that is, if it was required from the emergency room, mortality reached 100%. 

Another important goal in the treatment of TBI is the preservation of adequate oxygenation of the tissue. Indeed, if the severity of the injury does not allow for spontaneous oxygenation, invasive mechanical ventilation (IMV) could be required. Even though IMV usage could result in a reduction in secondary injuries, it could also be associated with an increased risk of some outcomes such as the development of pulmonary edema and infectious and/or aspiration pneumonia [12,56]. In this study, the use of IMV was recorded in 59% of the cases and was not associated with mortality or survival, which shows the severity of the clinical situation in this population. Previously, Taran et al. observed a 20% prevalence of IMV use in patients with TBI, with the majority being patients with polytrauma, therefore increasing mortality [57]. 

One of the most frequent sequelae of TBI in the short, medium and long term is seizures. There is a seven-times higher risk of presenting them and they are the most common cause of acquired epilepsy. The severity of the injury is a predisposing factor for this [38,58]. In the present study, prehospital seizures were included, while there was a low prevalence of these. When analyzing the survival curve, it was observed that having this condition reduced survival to 0%. A relationship with mortality was not observed; probably a larger sample number is required to present statistical significance and a positive relationship. Pease et al. and Hausted et al. found that convulsive seizures as sequelae influence the quality of life and costs of care in the health sector [58,59], which is not possible to compare since this study only focused on their presence or absence.

Even with an adequate hospital care protocol, the presence of brain death is common in patients with TBI; this increases if it is accompanied by hypotension at hospital admission, 24 h later and cerebral hemorrhage. These patients are potential candidates for organ donation [59]. According to Heppekcan et al., in their study of 245 patients, they observed that 26% were brain-dead, and of these, 26% were organ donors [60]. In this study, 9% had brain death and died within a period of 24 days. It is unknown if any of them were organ donors since this was not included as a variable, and only the association with mortality or survival was sought. However, it can be observed that there is a low incidence compared to other studies [59,60], either because of the sample size or the quality of medical care.

### Limitations

As with all retrospective studies, the present has its limitations. We found that the database had sampling bias and mistakes that could reduce the information of some variables, affecting the mortality analysis and its interpretation. Another point to improve is the presentation of the underreporting of some criteria that are relevant to this topic, such as toxicological screening tests for alcoholic intoxication and serum levels of illegal drugs, since they are not routinely performed in this hospital due to budget issues.

Also, in our research, the presence or absence of complications was quantified without describing, since the analyzed parameters are already quite extensive. From the perspectives of this study, some of this information could be reviewed in future projects.

## 5. Conclusions

TBI is an entity that has increased its incidence in recent years. It is preventable in the childbearing-age population if adequate protection measures are used and established traffic regulations are followed. Timely care impacts the productive years of life and reduces cost to the health department. Most cases in this population were of severe and moderate stages. The early identification of risk factors associated with mortality allows for treatment to be focused on preventing secondary injuries and improving the patient’s prognosis. Although most variables were not associated with mortality, the presence of seizures and the time of initiation of vasopressor medications did have an impact on survival, as well as the presence of trauma to the extremities as an additional injury. Blood pressure was close to significance similar to that reported in other studies conducted worldwide. All of the above encourages us to increase the sample size and avoid underreporting of data in future projects in order to observe whether there are changes in the statistical analysis.

## Figures and Tables

**Figure 1 biomedicines-12-02037-f001:**
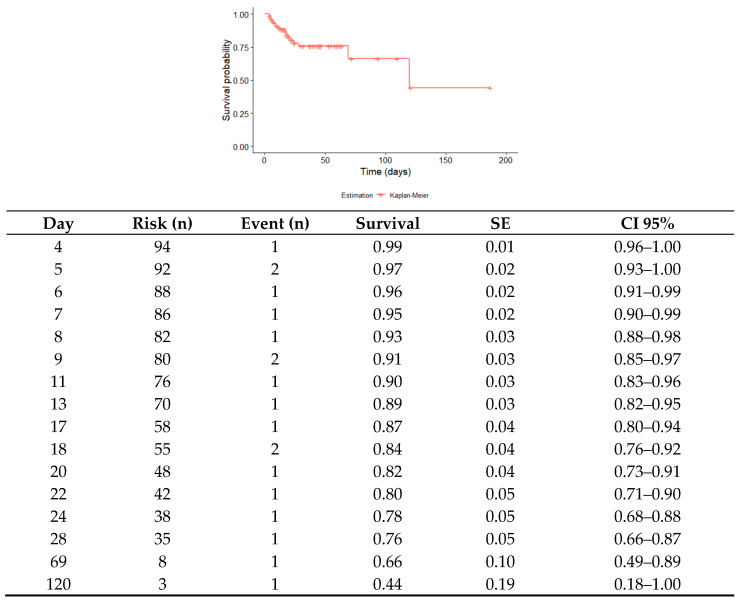
Survival curve in patients with TBI. A sample of 94 patients was studied. At the end of follow-up, there were 20 deaths, with a median survival of 120 days. The survival of the subjects at the end of the study was 0.44 (95% CI = 0.1, 1.0); data were modeled with the Kaplan–Meier estimation method. The survival and risk according to time, with corresponding 95% CI, are shown in the supplementary table.

**Figure 2 biomedicines-12-02037-f002:**
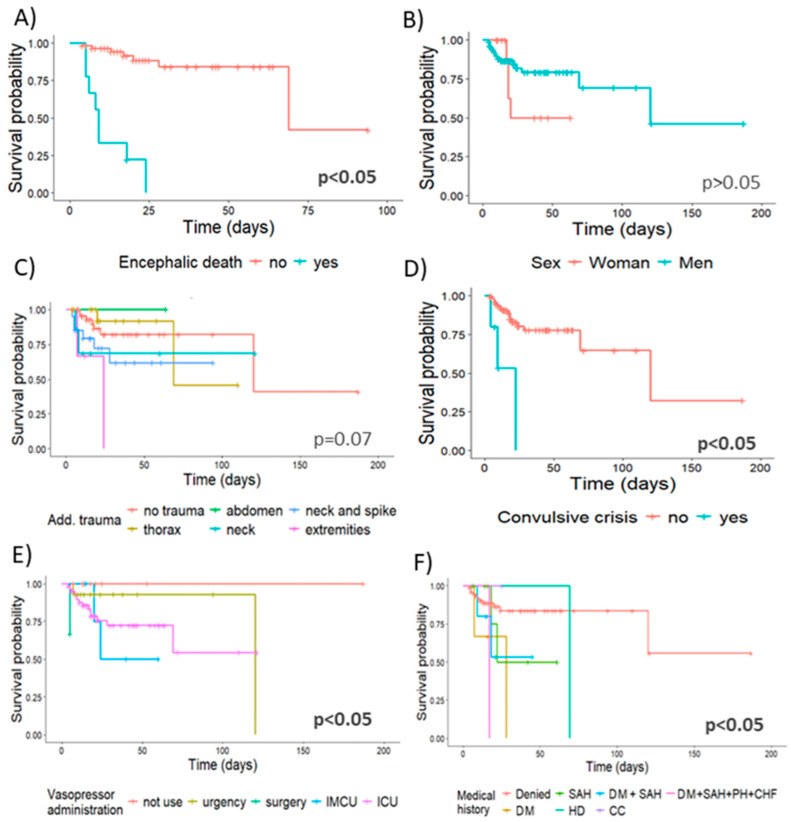
Risk factors for the survival of patients with TBI. A comparison of the survival curves of the most important risk factors for mortality identified in this study is presented. (**A**) Comparison by brain death, (**B**) comparison by sex, (**C**) additional trauma, (**D**) prehospital seizures, (**E**) use of vasopressors and (**F**) medical history. Survival curves for each risk factor were estimated using the Kaplan–Meier method. For comparisons of survival functions, they were performed with a log-rank test, and hypothesis tests were performed with a 95% CI. IMCU = Intermediate Care Unit, ICU = Intensive Care Unit, SAH = systemic arterial hypertension, DM = diabetes mellitus, PH = pulmonary hypertension, CHF = congestive heart failure, HD = heart disease and CC = cervical cancer.

**Table 1 biomedicines-12-02037-t001:** Clinical and demographic characteristics of patients with TBI.

	Frequency	Percentage	95% CI		Mean (S.D.)	95% CI
Mortality				Age (years)	37.81 (16.72)	34.4–41.2
Alive	75	77.30%	0.69–0.87	Weight (kg)	78.29 (14.18)	75.3–81.2
Deceased	19	19.60%	0.12–0.30	Initial GCS (points)	8.97 (3.67)	8.2–9.7
Sex				SBP emergencies (mmHg)	126.75 (25.81)	121.3–132.1
Men	82	87.20%	0.77–0.92	DBP emergencies (mmHg)	72.69 (14.24)	69.7–75.6
Women	12	12.70%	0.07–0.22	Vasopressor dosage (mcg/kg/min)	0.15 (0.12)	0.09–0.21
Influence of illegal drugs at time of injury				Initial Sh (g/dL)	13.24 (2.63)	12.6–13.8
Present	5	5.30%	0.01–0.11	Initial Sc (mg/dL)	1.02 (0.41)	0.9–1.1
Absent	89	95.60%	0.88–0.98	Hospital stays (days)	31.37 (30.49)	25.1–37.5
Influence of alcohol at time of injury				Hospital arrival time (days)	7.78 (31.37)	27.3–48.2
Present	22	23.40%	0.15–0.33			
Absent	72	76.50%	0.66–0.84			
Glasgow Coma Scale (points)						
Mild	21	22.34%	0.15–0.32			
Moderate	21	22.34%	0.15–0.33			
Severe	52	55.31%	0.43–0.64			
Trauma mechanism						
Physical aggression	6	6.30%	0.02–0.13			
Assault with weapon	8	8.50%	0.04–0.16			
Falls	31	32.90%	0.23–0.43			
Injured pedestrian	5	5.31%	0.01–0.12			
Car accident	11	11.70%	0.06–0.20			
Motorcycle accident	33	35.10%	0.25–0.45			

The clinical and demographic characteristics of patients with TBI that were measured upon admission to the medical emergency unit. The data for the categorical variables are presented as frequencies (Fr) and percentages, in addition to the 95% confidence intervals for proportions (95% CI). Data on quantitative variables are presented as means, standard deviation (SD) and 95% confidence intervals for the mean (95% CI). SBP = systolic blood pressure, DBP = diastolic blood pressure, Sh = serum hemoglobin, Sc = serum creatinine.

**Table 2 biomedicines-12-02037-t002:** Comparison of clinical characteristics according to mortality.

	Mean (SD)		*p*-Value	95% CI
	alive (n = 75)	deceased (n = 19)		
Age (years)	36.9 (15.23)	40.5 (19.3)	0.09	−13.54–6.28
Weight (kg)	78.7(14.21)	76 (14.81)	0.78	−5.47–10.91
SBP (mmHg)	125 (26.01)	131.6 (25.21)	0.61	−20.29–7
DBP (mmHg)	71.7 (15.04)	76 (11.09)	0.054	−11.8–3.23
Initial GCS (points)	9 (3.63)	9.3 (3.8)	0.77	−2.35–1.61
Vasopressor dosage (mcg/kg/min)	0.15 (0.13)	0.2 (0.05)	-	-
Sh (g/dL)	13.4 (2.72)	12.6 (1.98)	0.98	−0.6–2.19
Sc (mg/dL)	1.04 (0.42)	0.91 (0.11)	0.42	−0.03–0.29
Hospital stays (days)	33.1 (30.54)	21.7 (27.9)	0.35	−3.4–26.37

The data were analyzed with a Student’s t-test for data with a normal distribution and Mann–Whitney U for data that did not meet this statistical assumption. Statistically significant associations are assumed with a *p* < 0.05. In addition, the 95% confidence intervals for the difference in means (95% CI) are presented, as well as the mean and standard deviation. SBP = systolic blood pressure (emergency), DBP = diastolic blood pressure (emergency), Sh = initial serum hemoglobin, Sc = initial serum creatinine.

**Table 3 biomedicines-12-02037-t003:** Association of the clinical status of the patient with mortality.

	Alive	Deceased	*p*-Value	OR	95% CI
Sex					
Men	66	16	0.314	0.51	−1.9–0.7
Women	9	3			
Influence of illegal drugs at time of injury					
Present	4	1	0.817	1.31	−2.7–2.3
Absent	71	18			
Influence of alcohol at time of injury					
Present	18	4	0.765	0.82	−1.5–0.9
Absent	56	16			
Pupil state					
Isochoric	45	10	0.252	2.0	−0.5–1.8
Anisochoric	15	7	0.131	5.0	−0.6–3.8
Mydriasis	4	4	0.850	1.23	−2.8–2.2
Miosis	6	2	0.995	3.1 × 10^−7^	--
Enucleation	1	0	0.995	3.1 × 10^−7^	--
Pupillary reflex					
Reactive	5	1	0.824	0.77	−3.2–1.6
Unknown	70	18			
Complications					
Present	40	9	0.642	0.78	−1.2–0.7
Absent	35	10			
Use of vasopressors					
Present	21	3	0.283	0.48	−2.2–0.4
Absent	54	16			
Vasopressor type					
Norepinephrine	57	17	0.994	1.21 × 10^7^	--
Vasopressin	5	0	1.000	1.0	−98.8–98.8
Norepinephrine and vasopressin	10	5	0.994	1.59 × 10^7^	--
Use of mechanical ventilation					
Present	48	9	0.189	0.50	−1.7–0.3
Absent	27	10			
Use of hypertonic solution					
Present	16	5	0.663	1.29	−0.9–1.3
Absent	58	14			
Marshall scale on entry					
Diffuse type I lesion	4	2			
Diffuse type II lesion	12	2	0.404	0.27	−4.7–2.1
Diffuse type III lesion	19	11	0.675	1.66	−1.6–3.5
Diffuse type IV lesion	21	5	0.689	0.60	−2.8–2.6
Evacuated mass	4	0	0.991	1.91 × 10^−7^	--
Mass not evacuated	12	2	0.404	0.27	−4.7–2.1

The information was recorded upon admission of the patients to the emergency room, which was analyzed with a logistic regression, and statistically significant associations were assumed with a value of *p* < 0.05. In addition, the odds ratio (OR) values for each variable are presented with their respective 95% confidence intervals (95% CI).

**Table 4 biomedicines-12-02037-t004:** Risk factors for the survival of patients with TBI.

	HR	SE	CI 95%	*p*-Value
Encephalic-dead	17.48	0.584	5.56–54.93	***
Sex M	0.5111	0.573	0.16–1.57	0.242
Additional trauma				
thorax	0.70	0.806	0.15–3.73	0.745
abdomen	0	7860	0.00–inf	0.998
neck	1.98	0.828	0.39–10.07	0.407
neck and spine	2.45	0.562	0.81–7.39	0.110
extremities	7.11	0.813	1.44–35.01	0.015 *
Convulsive crisis	8.18	0.662	2.23–29.97	0.001 *
Vasopressor administration				
urgency	7.55 × 10^7^	8.04 × 10^3^	0–Inf	0.002 *
surgery	2.10 × 10^9^	8.04 × 10^3^	0–Inf	0.003 *
IMCU	2.55 × 10^8^	8.04 × 10^3^	0–Inf	0.002 *
ICU	1.66 × 10^8^	8.04 × 10^3^	0–Inf	0.002 *
Medical history				
DM	6.64	7.80 × 10^−1^	1.44–30.63	0.015 *
SAH	2.34	7.79 × 10^−1^	0.50–10.81	0.273
HD	3.51	1.09 × 10^0^	0.41–29.81	0.248
DM+SAH	3.56	7.81 × 10^−1^	0.77–16.5	0.103
CC	2.10 × 10^−7^	5.31 × 10^3^	0.00–inf	0.997
DM+SAH+PH+CHF	11.3	1.07 × 10^0^	1.38–92.91	0.023 *

Survival risk factors for patients with TBI. The hazard radius of the risk factors is presented with 95% CI, and statistical significance was assumed with a value *p* < 0.05 * and with a value *p* < 0.001 ***. IMCU = Intermediate Care Unit, ICU = Intensive Care Unit, SAH = systemic arterial hypertension, DM = diabetes mellitus, PH = pulmonary hypertension, CHF = congestive heart failure, HD = heart disease, and CC= cervical cancer.

**Table 5 biomedicines-12-02037-t005:** Brain death and seizures, i.e., the most important risk factors for mortality in subjects with TBI.

Cox’s Regression	Coefficient	Hazard Ratio	95% CI	SE (Coefficient)	*p*-Value
Encephalic death	2.589	13.3	3.90–45.47	0.626	***
Seizures	1.346	3.84	0.58–25.24	0.960	0.161
Age	−0.002	0.99	0.96–1.03	0.018	0.888

Survival analysis using the Cox regression method. The presented model corresponds to that with the greatest parsimony obtained, adjusted for the age of the traumatized subjects. *** *p* < 0.001, 95% CI of the hazard ratio, SE = standard error.

## Data Availability

The original contributions presented in the study are included in the article, further inquiries can be directed to the corresponding authors.

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
