# Peer review of "Mortality-Associated Factors in a Traumatic Brain Injury Population in Mexico"

_biomedicines, 2024, doi:10.3390/biomedicines12092037_

Round 1

Reviewer 1 Report

Comments and Suggestions for Authors

The article by Martinez-Herrera and colleagues aimed to identify the clinical and sociodemographic characteristics that influence the mortality or survival of 96 patients with TBI in a tertiary care hospital in Mexico. The article is of interest and important for the literature especially as real-world studies of TBI have been sparse in countries outside of the united states. However, there are many flaws in the interpretation of the data, the data can be represented better and need to be expanded upon, and the article needs to be checked for grammar and shortened. Because of this, I will say that the article can be reconsidered for publication after major revision. Below is a list of revisions that can occur in the first revision, after which the article will need to be reviewed again. 

I’m not sure that I agree with the main conclusion of the article that seizures and trauma aggregate in mortality in TBI patients. It appears there are multiple factors associated with mortality in this population. I think a better title for the paper would be something like, “Factors associated with mortality in a TBI population in Mexico.”

Did the authors receive informed consent from the research participants? There is also some concern regarding the fact that individuals below the age of 18 are included in this analysis. Can the authors explain more about their ethical approval process?

A Consort diagram should be used to detail inclusion and exclusion criteria for the study. 

Instead of stratifying patients by the terms “Alive” and “Dead” the authors should either just report on “mortality” (i.e. report only the “dead” patients with the assumption being that the rest are alive).

I would also suggest using the term “deceased” instead of “dead.” The term seems odd.

The authors use “Sex” to describe differences in males and females. Is this self-reported “sex”? If so, I belive this should be changed to Gender throughout the entire manuscript. The journal has reporting guidelines on this, which are briefly summarized here: Article titles and/or abstracts should indicate clearly what sex(es) the study applies to. Authors should also describe in the background, whether sex and/or gender differences may be expected; report how sex and/or gender were accounted for in the design of the study; provide disaggregated data by sex and/or gender, where appropriate; and discuss respective results.

Is Ethyl status alcoholic state? Can this please be explained in the methods. I believe this should be “under the influence of alcohol at time of injury” but this is hidden in the results section. Same with Drug Use. And drug use should be explained a bit more, what types of drugs are considered here?

First sentence of results state “A sample of 96 subjects with trauma was obtained…” but all figures only show 94 patients. What happened to the remaining 2 patients? A CONSORT diagram is needed.

Some information is missing from Table 1. In the text, the authors state, “The sample was mostly 139 made up from the lowest socioeconomic strata, level 1 represents 71.2% (95% CI= 59.88-80.54), level 2 22.5% (95% CI= 14.22-33.46), level 3 1.2% (95% CI= 0.06 - 7.72), level 4 2.5% (95% CI= 0.43 - 9.57) and level 5 2.5% (95% CI= 0.43 - 9.57), (See Table 1). Regarding marital status, 22.9% were married, 42.7% single, 26% lived in a common-law union, 2.1% were widowed, 1% were divorced and 5.2% were unknown.” But there is no information about socioeconomic status (SES) data shown in the table. Also, what is the meaning of these various levels of SES? Is this a standardized metric, and from where was it based? Last, I don’t think adding “Martial status” to the paper adds any value. This can be removed. 

Table 2, are the Glasgow scores reported correctly here? Individuals who died had a higher Glasgow score, or a slightly less severe brain injury? What do the authors make of this conundrum?

Table 1, top labels, shows “IC 95%.” should be “95% CI”. 

Table 1, Scholarship, “unknow” should be “Unknown”

Table 1, Figure legend, “The data for the categorical variables are presented as frequencies (Fr) and percentages, in addition to the 95% confidence intervals for proportions (95%CI1). Data on quantitative variables are presented as means, standard deviation (S.D.) and 95% confidence intervals for the mean (95%CI2). 1SBP=Systolic blood pressure, 2DBP=Diastolic blood pressure, 3Sh=serum hemoglobin, 4Sc=serum creatinine” The numbers after 95%CI1, 95%CI2 should be removed. And abbreviations aren’t need (e.g. (Fr, 1SBP, 2DBP, 3sh, 4SC, )) when they aren’t used. 

What’s the difference between “physical aggression” and “assault with a weapon”? Can assault with a weapon also be included in physical aggression?

Table 2, labels “alive” and “dead” are sometimes capitalized and sometimes not. Pick one. 

Figures 1 and 2, can the authors include number at risk in a table below the plots for each time point? Can the authors also include a Hazard ratio comparing the populations to a reference population when applicable?

Edits to Text:

Introduction, First sentence, “Traumatic brain injury (TBI) is an alteration of brain function….” this is confusing. A brain injury is the result of an external force that sometimes leads to alterations in brain function. Suggest rewrite this first sentence. 

Introduction, Second paragraph, second sentence, “According to the World Health Organization (WHO), 5.8 million people (10% 56 of global deaths) die each year.”  Suggest rewrite to “....die each year as a result of traumatic brain injuries.”

Introduction, Second paragraph, Among the causes, we find that a quarter are due to sui cides and homicides and another quarter because of traffic accidents (7). In America, traffic accidents are the leading cause of death in the ages of 15 to 29 (7,9). As for Mexico, 59 according to the National Institute of Statistics and Geography (INEGI), in 2022, deaths 60 due to accidents and aggressions placed in seventh and eighth respectively (8). It should 61 be noted, that in many cases, surviving patients suffer sequelae such as temporary and/or 62 permanent disabilities. There is an incidence of approximately 16% of posttraumatic dis- 63 abilities worldwide (7,9,10).” 

This should be rewritten. One suggestion is,  “Among the causes, according to (WHOEVER THE CITATION IS), a quarter of TBI-related deaths are due to suicides and homicides with another quarter resulting from traffic accidents. In the United States, trafffic accidents are the leading cause of TBI-related deaths in young adults (age 15 to 29 years old). In Mexico, however, according to a 2022 report from INEGI, deaths due to traffic accidents and violence were placed in seventh and eighth respectively [with XXXX [insert what it is] being the leading cause of TBI-related deaths in the country]. Additionally, for individuals who survive TBI events, patients often suffer from lifelong TBI-related impairments, with one report suggesting that 16% of TBI-survivors develop a permanent disability”

The third paragraph of the introduction has nothing to do with the study. It can be deleted completely. 

The discussion should be rewritten entirely to focus succinctly on the topic. It is much too long and needs to be more focused on the point.

Comments on the Quality of English Language

The quality of the English is flawed in the article, mostly in terms of grammar. I did not do extensive editing here, as I believe much of the article needs to be rewritten. Please focus on cutting out unnecessary words, sentences, and paragraphs. 

Author Response

Reviewer’s comments, author responses, and manuscript changes. The referee’s comments are in italics, the authors responses are in blue and the changes in the manuscript in yellow.

Answers to Reviewer 1 concerns:

The article by Martinez-Herrera and colleagues aimed to identify the clinical and sociodemographic characteristics that influence the mortality or survival of 96 patients with TBI in a tertiary care hospital in Mexico. The article is of interest and important for the literature especially as real-world studies of TBI have been sparse in countries outside of the United States.

Response= Thank you for your valuable comments an observations

Comments to the Author:

However, there are many flaws in the interpretation of the data, the data can be represented better and need to be expanded upon, and the article needs to be checked for grammar and shortened. Because of this, I will say that the article can be reconsidered for publication after major revision. Below is a list of revisions that can occur in the first revision, after which the article will need to be reviewed again.

  1. I’m not sure that I agree with the main conclusion of the article that seizures and trauma aggregate in mortality in TBI patients. It appears there are multiple factors associated with mortality in this population. I think a better title for the paper would be something like, “Factors associated with mortality in a TBI population in Mexico.”

Response= Thank you for your valuable comments. We have changed the title of the article to a more proper name “Factors associated with mortality in a TBI Mexican population

  1. Did the authors receive informed consent from the research participants? There is also some concern regarding the fact that individuals below the age of 18 are included in this analysis. Can the authors explain more about their ethical approval process?

Response= Thank you for the opportunity to clarify the following points. We want to let you know that for all patients one family member have signed an informed consent upon hospital admission, which includes authorization to all the procedures needed and for the use the data in research protocols performed at the hospital. Regarding patients under 17 years of age, there were 2 and they were eliminated from the project, therefore the initial sample was 96 and the one analyzed was 94 We have changed this information in the article to 94 patients.

In addition, all research projects are sent to the Research Ethics, Research and Biosafety committees, where they are evaluated by peers to analyze the research methodology, ethics, and all the important aspects of them. Then, the comments of the reviewers are sent to the Principal Investigator to be addressed. When the protocol has amendments and recommendations from the reviewers are sent for a second review or successive revisions until the protocol is ready for implementation

  1. A Consort diagram should be used to detail inclusion and exclusion criteria for the study. 

Response= Thank you for your valuable observations. “CONSORT (Consolidated Standards of Reporting Trials) is a protocol developed by a group of researchers not only to identify problems arising from conducting randomized controlled trials(RCTs), but also to report, in a full and clear manner, the results yielded by research, thereby facilitating RCTs reading and quality assessment” https://www.ncbi.nlm.nih.gov/pmc/articles/PMC4520133/#:~:text=In%20addition%20to%20the%20checklist,the%20RCT%20(Fig%201). Nevertheless, this protocol is for RCTs and the present is an epidemiological study without differences in the interventions of the patients not been appropriate the CONSORT flow diagram. It was rewritten the 2. Materials and Methods section for a better understanding of the participants selection as follows:

In the study, the sample of 94 patients comprised men and women aged 18 years and older and was determined with the following inclusion criteria: admitting diagnosis of mild to severe head trauma with initial prehospital treatment (use of anticonvulsants, crystal solutions or colloids, and mannitol), directly hospitalized from the emergency department or referred from other hospital units, with informed consent upon admission to hospital and from the emergency department, which included authorization for the observational study protocol, between February to October 2023. Exclusion criteria were polytrauma patients without head trauma. The elimination criteria were patients whose outcome was unknown, either by voluntary discharge or by transfer to another unit. The data were extracted from the electronic clinical record of the Ixtapaluca Regional High Specialty Hospital (HRAEI) and captured in an electronic platform called Single Registry of Patients with TBI created by the HRAEI planning service. The study was conducted in accordance with the Declaration of Helsinki and approved by the Research and Research Ethics Committees of the HRAEI (protocol code NR-32-2021 and NR-CEI-HRAEI-32-2021).

This is an observational, longitudinal, retrospective cohort study. The statistical analysis was carried out with the IBM SPSS version 25 program. Normality tests such as Kolmogorov-Smirnov, asymmetry, kurtosis, histogram and q-q graphs were performed to decide the corresponding statistical tests. For data with normal distribution, Student’s t-test and Pearson's test were applied, and for data with non-normal distribution we used Mann-Whitney U, Chi-square, Spearman's test, and risk estimation by odds ratio which was obtained through bivariate and multivariate logistic regression models. The survival curve and risk estimation were calculated using the Kaplan-Meier models.

  1. Instead of stratifying patients by the terms “Alive” and “Dead” the authors should either just report on “mortality” (i.e. report only the “dead” patients with the assumption being that the rest are alive). I would also suggest using the term “deceased” instead of “dead.” The term seems odd.

Response= Thanks for the observation, we have changed the terms.

  1. The authors use “Sex” to describe differences in males and females. Is this self-reported “sex”? If so, I believe this should be changed to Gender throughout the entire manuscript. The journal has reporting guidelines on this, which are briefly summarized here: Article titles and/or abstracts should indicate clearly what sex(es) the study applies to. Authors should also describe in the background, whether sex and/or gender differences may be expected; report how sex and/or gender were accounted for in the design of the study; provide disaggregated data by sex and/or gender, where appropriate; and discuss respective results.

Response= Thank you for your valuable comments. In this study, because it is retrospective and in accordance with the journal's guidelines, we only included sex (biological attribute) since in the electronic file the specification of gender (shaped by social and cultural circumstances) is not mandatory and therefore, it is not described in the usual way.

  1. Is Ethyl status alcoholic state? Can this please be explained in the methods. I believe this should be “under the influence of alcohol at time of injury” but this is hidden in the results section. Same with Drug Use. And drug use should be explained a bit more, what types of drugs are considered here?

Response= Thank you for your highly appreciate recommendation. We have changed the wording in accordance with the suggestion.

  1. First sentence of results state “A sample of 96 subjects with trauma was obtained…” but all figures only show 94 patients. What happened to the remaining 2 patients? A CONSORT diagram is needed.

Response= Thank you for your valuable observations. “CONSORT (Consolidated Standards of Reporting Trials) is a protocol developed by a group of researchers not only to identify problems arising from conducting randomized controlled trials(RCTs), but also to report, in a full and clear manner, the results yielded by research, thereby facilitating RCTs reading and quality assessment” https://www.ncbi.nlm.nih.gov/pmc/articles/PMC4520133/#:~:text=In%20addition%20to%20the%20checklist,the%20RCT%20(Fig%201). Nevertheless, this protocol is for RCTs and the present is an epidemiological study without differences in the interventions of the patients not been appropriate the CONSORT flow diagram. It was rewritten the 2. Materials and Methods section for a better understanding of the participants selection as follows:

In the study, the sample of 94 patients comprised men and women aged 18 years and older and was determined with the following inclusion criteria: admitting diagnosis of mild to severe head trauma with initial prehospital treatment (use of anticonvulsants, crystal solutions or colloids, and mannitol), directly hospitalized from the emergency department or referred from other hospital units, with informed consent upon admission to hospital and from the emergency department, which included authorization for the observational study protocol, between February to October 2023. Exclusion criteria were polytrauma patients without head trauma. The elimination criteria were patients whose outcome was unknown, either by voluntary discharge or by transfer to another unit. The data were extracted from the electronic clinical record of the Ixtapaluca Regional High Specialty Hospital (HRAEI) and captured in an electronic platform called Single Registry of Patients with TBI created by the HRAEI planning service. The study was conducted in accordance with the Declaration of Helsinki and approved by the Research and Research Ethics Committees of the HRAEI (protocol code NR-32-2021 and NR-CEI-HRAEI-32-2021).

This is an observational, longitudinal, retrospective cohort study. The statistical analysis was carried out with the IBM SPSS version 25 program. Normality tests such as Kolmogorov-Smirnov, asymmetry, kurtosis, histogram and q-q graphs were performed to decide the corresponding statistical tests. For data with normal distribution, Student’s t-test and Pearson's test were applied, and for data with non-normal distribution we used Mann-Whitney U, Chi-square, Spearman's test, and risk estimation by odds ratio which was obtained through bivariate and multivariate logistic regression models. The survival curve and risk estimation were calculated using the Kaplan-Meier models.

  1. Some information is missing from Table 1. In the text, the authors state, “The sample was mostly 139 made up from the lowest socioeconomic strata, level 1 represents 71.2% (95% CI= 59.88-80.54), level 2 22.5% (95% CI= 14.22-33.46), level 3 1.2% (95% CI= 0.06 - 7.72), level 4 2.5% (95% CI= 0.43 - 9.57) and level 5 2.5% (95% CI= 0.43 - 9.57), (See Table 1). Regarding marital status, 22.9% were married, 42.7% single, 26% lived in a common-law union, 2.1% were widowed, 1% were divorced and 5.2% were unknown.” But there is no information about socioeconomic status (SES) data shown in the table. Also, what is the meaning of these various levels of SES? Is this a standardized metric, and from where was it based? Last, I don’t think adding “Martial status” to the paper adds any value. This can be removed.

Response= Thank you for the opportunity to clarify the following pointstags. Sorry for missing the information of socioeconomical status. Indeed, it is a standardized metric and is regulated by the Ministry of Health, which is our health control body, issued on 05/27/2013 in "AGREEMENT by which the general criteria and methodology to which the socioeconomic classification processes of patients in the establishments that provide medical care services of the Ministry of Health and the entities coordinated by said Ministry must be subject to are issued," chapter FOUR (https://www.dof.gob.mx/nota_detalle.php?codigo=5300256&fecha=27/05/2013#gsc.tab=0). The classification indicates that level 1x and 1 are the strata with the least economic resources and stratum 6 is the stratum with the greatest economic resources. This classification is determined by a series of calculations that are explained in chapter ONE, TWO AND THREE of the mentioned law. In this study, the extreme strata are not observed, since there were no patients with these characteristics.

Last, I don’t think adding “Martial status” to the paper adds any value. This can be removed.

We agree with your valuable opinion, and it was eliminated as you rightly indicated.

  1. Table 2, are the Glasgow scores reported correctly here? Individuals who died had a higher Glasgow score, or a slightly less severe brain injury? What do the authors make of this conundrum?

Response= Thank you very much for your interesting and valuable observation In response to your comment, as stated in the first paragraph of the results "The average travel time from the accident site to the hospital was 37.8 minutes (95% CI= 27.3-48.2)." This is because HRAEI is the only third level hospital in the state (it has intensive care and specialists such as intensive care physicians, neurology or neurosurgery) and it has the ability to provide specialized and comprehensive care to these patients. Therefore, we are the first reference center for these patients and the neurological symptoms may not be fully present during this transfer time. In patients with longer transfer times, they are those who are received from other hospital units far from the HRAEI. The patients transferred by the other hospitals developed TBI far from the HRAEI but close to the other hospital units and that is why they are the first to hospitalize the patient. In this way, it is ensured that the transfer time is as short as possible and when a state of health suitable for transfer is achieved, they are received at the HRAEI. In these patients, the Glasgow score date of admission is taken from the clinical record of the other hospital unit.

In addition, as shown in Table 3, according to Marshall's classification, the most frequent injuries were extensive hemorrhages, and as we know, the neurological symptoms in intracranial hemorrhages are delayed.

  1. Table 1, top labels, shows “IC 95%.” should be “95% CI”. 

Response= Thank you for your appreciate observation. We have changed

  1. Table 1, Scholarship, “unknow” should be “Unknown”

Response= Thank you for your kind observation. We have changed it

  1. Table 1, Figure legend, “The data for the categorical variables are presented as frequencies (Fr) and percentages, in addition to the 95% confidence intervals for proportions (95%CI1). Data on quantitative variables are presented as means, standard deviation (S.D.) and 95% confidence intervals for the mean (95%CI2). 1SBP=Systolic blood pressure, 2DBP=Diastolic blood pressure, 3Sh=serum hemoglobin, 4Sc=serum creatinine” The numbers after 95%CI1, 95%CI2 should be removed. And abbreviations aren’t need (e.g. (Fr, 1SBP, 2DBP, 3sh, 4SC, )) when they aren’t used. 

Response= Thank you for your considerate observation. We have changed the information

  1. What’s the difference between “physical aggression” and “assault with a weapon”? Can assault with a weapon also be included in physical aggression?

Response= Thank you for the opportunity to explain the following points. In the case of physical aggression, it refers to injuries caused by physical force, blows from person to person. And in the case of assault with a weapon, it is the damage induced by the weapon itself.

  1. Table 2, labels “alive” and “dead” are sometimes capitalized and sometimes not. Pick one

Response= Thank you for your valuable observation. We have changed

  1. Figures 1 and 2, can the authors include number at risk in a table below the plots for each time point? Can the authors also include a Hazard ratio comparing the populations to a reference population when applicable?

Response= Thank you very much for your valuable comment. We have already included the tables as you suggested. However, for Figure 2, it is difficult to break down all the information completely in the paper, so a summary table has been inserted, and here we show the broken-down tables.

ENCEPHALIC DEATH

Without encephalic death

Days  risk(n) event(n) survival   S.E   lower95%CI   upper95%CI

     4     58       1    0.983  0.0171        0.950        1.000

     7     54       1    0.965  0.0246        0.917        1.000

   13     44       1    0.943  0.0324        0.881        1.000

   17     35       1    0.916  0.0412        0.838        1.000

   20     29       1    0.884  0.0504        0.791        0.989

   28     21       1    0.842  0.0632        0.727        0.975

   69       2       1    0.421  0.2994        0.104        1.000

Encephalic death

Days  risk(n) event(n) survival   S.E   lower95%CI   upper95%CI

    5      9       2    0.778   0.139       0.5485        1.000

    6      7       1    0.667   0.157       0.4200        1.000

    8      6       1    0.556   0.166       0.3097        0.997

    9      5       2    0.333   0.157       0.1323        0.840

   18     3       1    0.222   0.139       0.0655        0.754

   24     1       1    0.000    NaN           NA           NA

SEX DIFFERENCES

Women

Days  risk(n) event(n) survival   S.E   lower95%CI   upper95%CI

    17      8       1     0.875   0.117        0.673        1.000

    18      7       2     0.625   0.171        0.365        1.000

    20      5       1     0.500   0.177        0.250        1.000

Men

Days  risk(n) event(n) survival   S.E   lower95%CI   upper95%CI

    4      81       1    0.988  0.0123        0.964        1.000

    5      79       2    0.963  0.0212        0.922        1.000

    6      75       1    0.950  0.0245        0.903        0.999

    7      73       1    0.937  0.0274        0.885        0.992

    8      69       1    0.923  0.0302        0.866        0.984

    9      67       2    0.896  0.0350        0.830        0.967

   11     64       1    0.882  0.0371        0.812        0.958

   13     59       1    0.867  0.0394        0.793        0.947

   22     38       1    0.844  0.0445        0.761        0.936

   24     34       1    0.819  0.0496        0.727        0.922

   28     31       1    0.793  0.0546        0.693        0.907

   69      8        1    0.694  0.1043        0.517        0.931

 120      3        1    0.462  0.2012        0.197        1.000

TRAUMA

Without trauma

Days  risk(n) event(n) survival   S.E   lower95%CI   upper95%CI

     9     43       2    0.953  0.0321        0.893        1.000

   13     37       1    0.928  0.0403        0.852        1.000

   17     29       1    0.896  0.0500        0.803        0.999

   18     27       1    0.863  0.0581        0.756        0.984

   22     21       1    0.821  0.0683        0.698        0.967

  120      2       1    0.411  0.2924        0.102        1.000

thorax

Days  risk(n) event(n) survival   S.E   lower95%CI   upper95%CI

   20     12       1    0.917  0.0798        0.773        1.000

   69       2       1    0.458  0.3265        0.113        1.000

neck

Days  risk(n) event(n) survival   S.E   lower95%CI   upper95%CI

    6        7       1    0.857   0.132        0.633         1.000

    8        5       1    0.686   0.186        0.403         1.000

neck and spike

Days  risk(n) event(n) survival   S.E   lower95%CI   upper95%CI

    4     20       1    0.950  0.0487        0.859        1.000

    5     19       2    0.850  0.0798        0.707        1.000

   11    15       1    0.793  0.0925        0.631        0.997

   18    11       1    0.721  0.1086        0.537        0.969

   28      7       1    0.618  0.1333        0.405        0.943

extremities

Days  risk(n) event(n) survival   S.E   lower95%CI   upper95%CI

    7      3       1    0.667    0.272          0.3            1.000

  24      1       1    0.000    NaN           NA           NA

CONVULSIVE CRISIS

Without convulsive crisis

Days  risk(n) event(n) survival   S.E   lower95%CI   upper95%CI

     5      79       1    0.987  0.0126       0.9630        1.000

     6      76       1    0.974  0.0179       0.9399        1.000

     7      75       1    0.961  0.0219       0.9194        1.000

     8      72       1    0.948  0.0253       0.8997        0.999

     9      70       1    0.934  0.0284       0.8805        0.992

    11     68       1    0.921  0.0311       0.8618        0.984

    13     62       1    0.906  0.0339       0.8417        0.975

    17     50       1    0.888  0.0378       0.8167        0.965

    18     47       2    0.850  0.0446       0.7668        0.942

    20     40       1    0.829  0.0483       0.7392        0.929

    24     33       1    0.804  0.0530       0.7062        0.914

    28     30       1    0.777  0.0576       0.6718        0.898

    69       6       1    0.647  0.1276       0.4400        0.953

  120       2       1    0.324  0.2376       0.0768        1.000

Convulsive crisis

Days  risk(n) event(n) survival   S.E   lower95%CI   upper95%CI

    4      5       1    0.800   0.179        0.516            1

    9      3       1    0.533   0.248        0.214            1

   22      1       1    0.000     NaN           NA           NA

VASOPRESSORS

administration of vasopressors in EMERGENCIES

Days  risk(n) event(n) survival   S.E   lower95%CI   upper95%CI

    7     14       1    0.929  0.0688        0.803            1

  120      1       1    0.000     NaN           NA           NA

administration of vasopressors in Qx

Days  risk(n) event(n) survival   S.E   lower95%CI   upper95%CI

    5      3       1    0.667   0.272        0.300        1.000

administration of vasopressors in UTIA

Days  risk(n) event(n) survival   S.E   lower95%CI   upper95%CI

   20      4       1     0.75   0.217        0.426            1

   24      3       1     0.50   0.250        0.188            1

administration of vasopressors in UCIA

Days  risk(n) event(n) survival   S.E   lower95%CI   upper95%CI

     4     59       1    0.983  0.0168        0.951        1.000

     5     57       1    0.966  0.0238        0.920        1.000

     6     56       1    0.949  0.0289        0.894        1.000

     8     52       1    0.930  0.0336        0.867        0.999

     9     51       2    0.894  0.0410        0.817        0.978

   11     49       1    0.876  0.0441        0.793        0.966

   13     45       1    0.856  0.0472        0.768        0.954

   17     38       1    0.834  0.0510        0.739        0.940

   18     35       2    0.786  0.0582        0.680        0.909

   22     27       1    0.757  0.0629        0.643        0.891

   28     23       1    0.724  0.0682        0.602        0.871

   69       4       1    0.543  0.1649        0.299        0.985

MEDICAL HISTORY

Denied

Days  risk(n) event(n) survival   S.E   lower95%CI   upper95%CI

     4     75       1    0.987  0.0132        0.961        1.000

     5     73       2    0.960  0.0228        0.916        1.000

     6     69       1    0.946  0.0264        0.895        0.999

     8     66       1    0.931  0.0296        0.875        0.991

     9     64       1    0.917  0.0325        0.855        0.983

   11     61       1    0.902  0.0353        0.835        0.974

   13     55       1    0.885  0.0383        0.813        0.964

   20     40       1    0.863  0.0433        0.783        0.952

   24     32       1    0.836  0.0496        0.745        0.939

  120      3       1    0.558  0.2300        0.248        1.000

Mellitus diabetes (MD)

Days  risk(n) event(n) survival   S.E   lower95%CI   upper95%CI

    7      3       1    0.667   0.272          0.3            1

   28      1       1    0.000     NaN           NA           NA

Systemic arterial hypertension (SAH)

Days  risk(n) event(n) survival   S.E   lower95%CI   upper95%CI

   18      4       1     0.75   0.217        0.426            1

   22      3       1     0.50   0.250        0.188            1

Heart Disease (HD)

Days  risk(n) event(n) survival   S.E   lower95%CI   upper95%CI

   69      1       1        0     NaN           NA           NA

MD+SAH

Days  risk(n) event(n) survival   S.E   lower95%CI   upper95%CI

    9      5       1    0.800   0.179        0.516            1

   18      3       1    0.533   0.248        0.214            1

MD+SAH+PH+ICC

Days  risk(n) event(n) survival   S.E   lower95%CI   upper95%CI

   17      1       1    0         NaN           NA           NA

Edits to Text:

  1. Introduction, First sentence, “Traumatic brain injury (TBI) is an alteration of brain function….” this is confusing. A brain injury is the result of an external force that sometimes leads to alterations in brain function. Suggest rewrite this first sentence. 

Response= Thank you for your comments, this sentence is a frequently used definition, the following examples are attached:

Hassett, et al. (2023) “Traumatic brain injury (TBI) is defined as an alteration in brain function caused by an external force to the head, such as traffic accidents (car, bicycle, pedestrian), falls, blast injuries, acts of violence, and sports injuries.”

Menon, D. K., et al. (2010) “TBI is defined as an alteration in brain function, or other evidence of brain pathology, caused by an external force.”

Soto-Páramo, et al. (2022) "Traumatic brain injury (TBI) is defined as a temporary or permanent alteration in brain function caused by an external force. . ."

  1. Hassett, L. Physiotherapy management of moderate-to-severe traumatic brain injury. J Physiother 2023, 69, 141-47. https://doi.org/10.1016/j.jphys.2023.05.015
  2. Menon, D.K.; Schwab, K.; Wright, D.W.; Maas, A.I. Position Statement: Definition of Traumatic Brain Injury. Arch Phys Med Rehabil 2010,91,1637–40. https://doi.org/10.1016/j.apmr.2010.05.017
  3. Soto-Páramo, D.G.; Pérez-Nieto, O.R.; Deloya-Tomas, E.; Rayo-Rodríguez, S.; Castillo-Gutiérrez, G.; Olvera-Ramos, M.G.; et al. Pathophysiology, diagnosis and treatment of traumatic brain injury. Neurologia, Neurocirugia y Psiquiatria 2022, 50, 4–15.

For this reason, we consider it prudent to respect the definition.

  1. Introduction, Second paragraph, second sentence, “According to the World Health Organization (WHO), 5.8 million people (10% 56 of global deaths) die each year.” Suggest rewrite to “....die each year as a result of traumatic brain injuries.”

Response= Thank you for your comment, it was done as you indicated.

  1. Introduction, Second paragraph, “Among the causes, we find that a quarter are due to sui cides and homicides and another quarter because of traffic accidents (7). In America, traffic accidents are the leading cause of death in the ages of 15 to 29 (7,9). As for Mexico, 59 according to the National Institute of Statistics and Geography (INEGI), in 2022, deaths 60 due to accidents and aggressions placed in seventh and eighth respectively (8). It should 61 be noted, that in many cases, surviving patients suffer sequelae such as temporary and/or 62 permanent disabilities. There is an incidence of approximately 16% of posttraumatic dis- 63 abilities worldwide (7,9,10).” This should be rewritten. One suggestion is,  “Among the causes, according to (WHOEVER THE CITATION IS), a quarter of TBI-related deaths are due to suicides and homicides with another quarter resulting from traffic accidents. In the United States, trafffic accidents are the leading cause of TBI-related deaths in young adults (age 15 to 29 years old). In Mexico, however, according to a 2022 report from INEGI, deaths due to traffic accidents and violence were placed in seventh and eighth respectively [with XXXX [insert what it is] being the leading cause of TBI-related deaths in the country]. Additionally, for individuals who survive TBI events, patients often suffer from lifelong TBI-related impairments, with one report suggesting that 16% of TBI-survivors develop a permanent disability”

Response= Thank you very much for your valuable suggestion. We really think that the suggested wording was the best, so we included it.

  1. The third paragraph of the introduction has nothing to do with the study. It can be deleted completely.

Response= Thank you for your comment, we have changed the wording.

  1. The discussion should be rewritten entirely to focus succinctly on the topic. It is much too long and needs to be more focused on the point.

Response= Thank you for your suggestion, we have already removed from the text everything that was not relevant in the section 4. Discussion

Comments on the Quality of English Language: The quality of the English is flawed in the article, mostly in terms of grammar. I did not do extensive editing here, as I believe much of the article needs to be rewritten. Please focus on cutting out unnecessary words, sentences, and paragraphs.

Response= Thank you for kind remarkable statement. The text has been reviewed by a native speaker and the modifications have been made in the mentioned terms, as well as the English wording has been improved.

Reviewer 2 Report

Comments and Suggestions for Authors

I reviewed carefully the manuscript title: “Seizures and trauma aggregate in mortality associated with traumatic brain injury”, however, major and minor modifications are needed as follows:

Major

Abstract. Result section add the number of patients analyzed.

In methods, as was defined the pre-hospital treatment.

Results: The line 124: with a mortality 124 rate of 77.3% (95% CI= 0.69-0.87) in incorrect because is 77.3% alive as is mentioned in the Table 1.  

On line 142 the marital status is irrelevant in this study.

Table 2 show redundancy with the lines 147-153.

On table 3 the variable complications are systemic or neurological complications or both.

On the discussion one limitation is the lack of toxicological screening test in all the patients. In line 347 this study has the same limitation as well.

The lines of the conclusion 394-396 are not supported by the results also what is the relevance to include non significant results in this sectionBlood pressure was close to the significance similar to that reported in other studies conducted worldwide. The same problem remain in the conclusion of the abstract without any mention about the seizures and other relevant findings.

Minor

In Table 1 and 3. Illegal drug use? Or what kind of drugs are considered. Also change Ethyl state for alcoholic intoxication term and add the full name of Glasgow Score Scale. Additional all the units of measurements of multiples variables are missing and repeated with the text, avoid redundance in the text. For example 135-137 lines

Use your TBI abbreviation in the text for example line 146

In table 2 add the SD to the mean column using (  ).

Line 197 change spike for spine

On Table 4, change Convulsive crisis to only seizures

Comments on the Quality of English Language

Mentioned previously

Author Response

Reviewer’s comments, author responses, and manuscript changes. The referee’s comments are in italics, the authors responses are in blue and the changes in the manuscript in yellow.

Answers to Reviewer 2 concerns:

I reviewed carefully the manuscript title: “Seizures and trauma aggregate in mortality associated with traumatic brain injury”, however, major and minor modifications are needed as follows:

Response= Thank you for your considerate comments an observations

Comments to the Author:

  1. Major
  1. Abstract. Result section add the number of patients analyzed.

Response= Thank you for your valuable remark. We have added the information.

  1. In methods, as was defined the pre-hospital treatment.

Response= Thank you for your helpful comment. In the 2. Materials and Methods section has already specified what we mean by pre-hospital treatment as follows:

  1. Materials and Methods

In the study, the sample of 94 patients comprised men and women aged 18 years and older and was determined with the following inclusion criteria: admitting diagnosis of mild to severe head trauma with initial prehospital treatment (use of anticonvulsants, crystal solutions or colloids, and mannitol), directly hospitalized from the emergency department or referred from other hospital units, with informed consent upon admission to hospital and from the emergency department, which included authorization for the observational study protocol, between February to October 2023. Exclusion criteria were polytrauma patients without head trauma. The elimination criteria were patients whose outcome was unknown, either by voluntary discharge or by transfer to another unit. The data were extracted from the electronic clinical record of the Ixtapaluca Regional High Specialty Hospital (HRAEI) and captured in an electronic platform called Single Registry of Patients with TBI created by the HRAEI planning service. The study was conducted in accordance with the Declaration of Helsinki and approved by the Research and Research Ethics Committees of the HRAEI (protocol code NR-32-2021 and NR-CEI-HRAEI-32-2021).

This is an observational, longitudinal, retrospective cohort study. The statistical analysis was carried out with the IBM SPSS version 25 program. Normality tests such as Kolmogorov-Smirnov, asymmetry, kurtosis, histogram and q-q graphs were performed to decide the corresponding statistical tests. For data with normal distribution, Student’s t-test and Pearson's test were applied, and for data with non-normal distribution we used Mann-Whitney U, Chi-square, Spearman's test, and risk estimation by odds ratio which was obtained through bivariate and multivariate logistic regression models. The survival curve and risk estimation were calculated using the Kaplan-Meier models.

  1. Results: The line 124: with a mortality 124 rate of 77.3% (95% CI= 0.69-0.87) in incorrect because is 77.3% alive as is mentioned in the Table 1.

Response= Thank you for your valuable remark. We have changed

  1. On line 142 the marital status is irrelevant in this study.

Response= Thank you for the observation. We have eliminated the marital status analysis.

  1. Table 2 show redundancy with the lines 147-153.

Response= Thank you for your comment, lines 147 to 150 are the footer of the table, and lines 151 to 153 were changed as follows: "Regarding the association of the clinical status of traumatized subjects with mortality, only age and BPD could be risk factors." We believe it is important not to remove that sentence, as it emphasizes the significant results.

  1. On table 3 the variable complications are systemic or neurological complications or both.

Response= In this study, we only measured the presence or absence of complications, since the parameters we analyzed are already quite extensive and we prefer to expand this information in future projects.

  1. On the discussion one limitation is the lack of toxicological screening test in all the patients. In line 347 this study has the same limitation as well.

Response= Thank you for your remarkable observation. We have added more specific parameters in the Limitations section

  1. The lines of the conclusion 394-396 are not supported by the results also what is the relevance to include non-significant results in this section Blood pressure was close to the significance similar to that reported in other studies conducted worldwide. The same problem remain in the conclusion of the abstract without any mention about the seizures and other relevant findings.

Response= Thank you very much for your comment. The wording in all the manuscript has been corrected to emphasize the key points of this study.

  1. Minor
  1. In Table 1 and 3. Illegal drug use? Or what kind of drugs are considered. Also change Ethyl state for alcoholic intoxication term and add the full name of Glasgow Score Scale. Additional all the units of measurements of multiples variables are missing and repeated with the text, avoid redundance in the text. For example 135-137 lines

Response= Thank you for your valuable comment, the relevant changes have already been made to the text and tables. And with drugs, if we are referring to illegal drugs, we do not have the data on the type of illegal drug since it is not specified in the clinical record.

  1. Use your TBI abbreviation in the text for example line 146

Response= Thank you for the suggestion, we have corrected.

  1. In table 2 add the SD to the mean column using (  ).

Response= Thank you for your comment, it was done the suggestion as follows:

Table 2. Comparison of clinical characteristics according to mortality.

Mean (S.D.)

p-value

95%CI

alive (n=75)

deceased (n=19)

Age

36.9 (15.23)

40.5 (19.3)

0.09

-13.54 – 6.28

Weight

78.7(14.21)

76 (14.81)

0.78

-5.47 – 10.91

SBP

125 (26.01)

131.6 (25.21)

0.61

-20.29 – 7

DBP

71.7 (15.04)

76 (11.09)

0.054

-11.8 – 3.23

Initial GCS

9 (3.63)

9.3 (3.8)

0.77

-2.35 – 1.61

Vasopressor dosage

0.15 (0.13)

0.2 (0.05)

-

-

Sh

13.4 (2.72)

12.6 (1.98)

0.98

-0.6 – 2.19

Sc

1.04 (0.42)

0.91 (0.11)

0.42

-0.03 – 0.29

Hospital stays

33.1 (30.54)

21.7 (27.9)

0.35

-3.4 – 26.37

Table 2. The data were analyzed with a Student t test for data with a normal distribution and Mann-Whitney U for data that do not meet this statistical assumption. Statistically significant associations are assumed with a p<0.05*; In addition, the 95% confidence intervals for the difference in means (95%CI) are presented, as well as the mean and standard deviation. SBP= systolic blood pressure (emergency), DBP= diastolic blood pressure (emergency), Sh= inicial serum hemoglobin, Sc= inicial serum creatinine.

  1. Line 197 change spike for spine

Response= Thank you for the kind observation, we have corrected.

  1. On Table 4, change Convulsive crisis to only seizures

Response= Thank you for the considerate statement, we have corrected.

Round 2

Reviewer 1 Report

Comments and Suggestions for Authors

The authors have adequately addressed most of my comments. I think the paper is acceptable for the journal. 

The paper should include more information about how the data was obtained. It should state somewhere (probably in methods) that the data came from self-reported and clinician-reported medical records.

There are still a few issues, for example in Table 1, Glasgow Coma Scale, the number of patients here add up to 96 instead of 94. 

Table 3, for all categories. The numbers don't add up to 75. In cases, where the information is unknown, the authors should include the category "unknown" with the appropriate number. for example, Influence of illegal drugs at time of injury, present = 3, absent = 71, unknown = 1.  AND vasopressor type, NE = 56, vasopressin = 2, NE +Vasopressin = 8, Unknown (or none) = 9. 

For  figure 2b, the p-value of  p > 0.05 should report the actual p-value instead of greater than 0.05. 

There are a couple places where Spanish is used instead of English, for example, table 5, "sexo M". Table 2, figure legend, "inicial"

Comments on the Quality of English Language

There are a few instances where Spanish is used in the text, but otherwise the language is fine. 

Reviewer 2 Report

Comments and Suggestions for Authors

Thanks so much for the modifications to the paper additionally include the unit of measurement of the variables in the tables.
